# Prevalence of Antibiotic Use and Disposal at Household Level in Informal Settlements of Kisumu, Kenya

**DOI:** 10.3390/ijerph20010287

**Published:** 2022-12-24

**Authors:** Kellen J. Karimi, Aijaz Ahmad, Adriano Duse, Mutuku Mwanthi, Richard Ayah

**Affiliations:** 1Clinical Microbiology and Infectious Diseases, School of Pathology, Faculty of Health Sciences, University of the Witwatersrand, Johannesburg 2193, South Africa; 2Department of Public and Global Health, Faculty of Health Sciences, University of Nairobi, Nairobi P.O. Box 19676-00202, Kenya; 3Infection Control, Charlotte Maxeke Johannesburg Academic Hospital, National Health Laboratory Service, Johannesburg 2193, South Africa

**Keywords:** antibiotic use, antibiotic disposal, human health effect, antibiotic resistance, environment, informal settlements

## Abstract

The use and abuse of antibiotics are directly related to the development of drug resistance, a global public health problem. Whereas the majority of research focus is on the use and misuse of antibiotics in drug resistance development, little is known about improper disposal, as a source of contamination in the environment that includes groundwater, especially in informal settlements. This study sought to determine antibiotic use and disposal in informal settlements in Kisumu, Kenya. A random cross-sectional sample of 447 households in selected informal settlements of Kisumu, Kenya was studied. A structured questionnaire was issued to persons heading households. The prevalence of antibiotic use was 43% (n = 193). Among these people, 74% (n = 144) had consulted a health worker in a healthcare facility for a prescription. Respondents did not always complete doses but kept the remainder for the next time they would become ill (54%). About 32% disposed of the remainder of the antibiotics in pit latrines and compost pits (10%) while 4% disposed through burning. Antibiotic use was fairly high despite a low level of awareness of the health effects of consuming water contaminated with antibiotics (35%) (n = 156); *p* = 0.03. Misuse and inappropriate disposal of antibiotics as identified may lead to a higher risk of antibiotic resistance, increasing the disease burden in the informal settlements.

## 1. Introduction

Bacterial infections are a leading cause of morbidity and mortality around the world. Antibiotics continue to be a low-cost solution to current human and animal health problems both in treatment and prophylaxis [1,2]. They improve health outcomes, along with improved water and sanitation, nutrition, and vaccination [3]. The positive gains, however, are offset by rising antimicrobial resistance, which is facilitated by increased antibiotic consumption, with many low- and middle-income countries lacking the capacity to conduct surveillance [3]. According to the World Health Organization, approximately 50% of antibiotics consumed are unnecessary [4], resulting in antibiotic effectiveness being compromised owing to resistance [5,6,7]. Individual patients’ antibiotic use is defined as when they receive the correct medicine for the appropriate indication, in doses that match their individual requirements for a specified period of time, and with the correct information [8]. Irrational antibiotic use, on the other hand, occurs when any of the predefined conditions are not met, or when the existing algorithm for determining the presence of infection among patients to warrant a prescription is not followed [6].

Antibiotic consumption increased by 39% globally between 2000 and 2015 [9]. Consumption rates were found to be high in North America, Europe, and the Middle East [3]. Consumption is lower in low- and middle-income countries, despite the fact that the bacterial disease burden that causes death is higher in low-income countries [10,11,12], but the LMICs are catching up and may soon register rates comparable to high-income countries [9]. Global consumption rates were reported to have increased by 46% overall [3]. 

Antibiotic use has been prevalent in hospital settings, where treatment guidelines recommend optimal antibiotic use in order to reduce the threat of antibiotic resistance, which worsens the burden of infectious diseases [12,13]. Antibiotic use has been reported in low- and middle-income countries, both in hospital settings [14] and in communities where misuse is common, with the majority preferring self-medication [15]. Misuse is largely fuelled by the ease of obtaining antibiotics without a prescription, a scenario described as less time-consuming, less expensive, and more convenient than visiting health facilities for a medical review [15]. In Kenya, an antibiotic consumption study was conducted in an informal settlement in Nairobi, where the prevalence of antibiotic use was found to be high (87% (initial survey) and 70% (follow-up survey)). This happened in addition to a lack of understanding about effective antibiotic use. [14].

The World Health Organization reported the prevalence of antibiotic consumption in defined daily dosages for four Sub-Saharan African countries: the Republic of Tanzania, Burkina Faso, Cote d’Ivoire, and Burundi, with defined daily dosages of 27.3, 13.8, 10.7, and 4.4 per 1000 inhabitants per day, respectively [16]. However, no data on antibiotic usage have been reported in Kenya, making it impossible to implement the set antibiotic use and antibiotic resistance guidelines [17]. 

Antibiotics are used for the treatment of infection and for prophylaxis among different populations [18], especially persons living with HIV patients, who use co-trimoxazole (sulfamethoxazole and trimethoprim) for prophylaxis and treatment of related infections [19]. For instance, among HIV patients, the World Health Organization recommends initiation of co-trimoxazole in patients with HIV clinical disease (stages 3 and 4) and/or a CD4+ cell count of ≤350 cells [19,20]. With the threat of antibiotic resistance, HIV patients are at risk of experiencing side effects that might lead to an increased burden of disease and death.

Antibiotic misuse has been categorized as both community- and healthcare-provider-driven owing to a lack of understanding and awareness of the dangers of antibiotic abuse in the community, availability of antibiotics without a prescription, and consumption of leftover medications, even though prescibers and dispensers were found to have adequate knowledge on antibiotic use [7,21]. The leftover antibiotics are available because patients are unable to complete the medication provided to them for a variety of reasons that vary from person to person, such as adverse effects, feeling better, or drug expiry [22,23]. Predictors of antibiotic misuse among healthcare providers include pharmacists’ and prescribers’ knowledge, attitude, and practice, patient/doctor contact, a lack of rapid diagnostic tests, pharmaceutical promotion, and a lack of adequate antibiotic education for healthcare professionals [7,21].

Environmental contamination with antibiotics has been linked to a number of sources, including pit latrines, sewage, wastewater, and animal and hospital waste disposal [24,25,26,27,28]. Other sources of groundwater contamination include industrial waste, agricultural runoff, and floods, which are distressing to the locals of these settlements studied and are linked to the shallow aquifer [29]. Antibiotics are becoming more and more prevalent in the environment, but it is unclear how they are used and disposed of in homes, given that they are present in water and what role they play in unintentional exposure [30]. To prevent environmental pollution, which fosters the development of antibiotic resistance, it is critical to properly and safely dispose of any unused drugs [18].

An increased urban population has resulted in the expansion of informal settlements, which has a negative influence on water supplies [31]. Residents of informal communities must consequently balance the scarcity with readily available groundwater by sinking shallow wells [32]. As the technology required to make water safe may not be readily available to informal settlement dwellers, groundwater may be consumed without being subjected to any type of treatment owing to its clean appearance. Residents who are unaware of the presence of antibiotics in their drinking water may be at risk of exposure.

Since there exists a complex interaction between humans, animals, and the environment [33], health protection in a contaminated environment is critical. This strategy will attempt to attain optimal health while also addressing the future global antibiotic resistance burden, particularly in informal settlements [34]. This paper examines antibiotic use and disposal in informal settlement households. 

## 2. Materials and Methods

The study was conducted in Kisumu County, which is located in Kenya’s western area, 350 km from the capital city of Nairobi, at longitudes 33°20’ E and latitudes 0°20’ S and 0°50’ S [35]. Kisumu has a total land area of 2086 km^2^, of which roughly 567 km^2^ is water. The county is located on the beaches of Lake Victoria, the world’s third-biggest freshwater lake, which also borders Kenya, Uganda, and Tanzania. The research sites are located in the Kisumu East and Kisumu Central sub-counties. There are eight informal settlements in the county, five of which were studied: Obunga, Manyatta A, Manyatta B, Nyalenda A, and Nyalenda B.

The population density of the informal settlements is projected to be 567,983 [36], with an annual urbanization rate of 2.8 percent [36]. The population in informal settlements relies on groundwater to supply their residential water needs while also using the sub-surface for sanitation facilities. The majority use pit latrines [32]. 

A cross-section of households in the chosen informal settlements was studied in September 2019. The AF-RIWATSAN project site, which included a number of components, was purposively selected [37]. Prior to the study, groundwater sources were mapped, and a sample of homes was selected at random, based on the population of each informal settlement around the identified groundwater sources [38]. A sample size of 442 households was determined. A proportionate distribution of homes was used, depending on the population of each of the five (5) informal settlements in the study area, namely Manyatta A and B, Nyalenda A and B, and Obunga, as reported by the Kenya Bureau of Statistics, to guarantee that each was sufficiently sampled. This ensured that more participants were drawn from more populous informal settlements, and vice versa. One individual from each home was chosen as the head to be involved in the study and interviewed after providing informed consent. The first household was chosen based on its proximity to the informal settlement’s administrative leadership, being the one on the right-hand side of the chief’s office. Every fifth family was chosen and included in the sample until the determined sample size [39] of 442 households was achieved. 

A structured questionnaire was administered by trained interviewers using the Swahili language which was understood by the respondents. To verify that the translation from English to Swahili was not lost, the validity of the questions was maintained, and the possibility for introducing information bias was reduced, the questionnaire was piloted among residents of a different informal community within Kisumu. Groundwater use and the use of antibiotics were outcome variables. Sociodemographic traits, the source of the antibiotics, the completion of the antibiotic dose, and the disposal of leftover drugs were all predictor variables. Version 20 of IBM* SPSS* Statistics was used to enter the data, and also conduct descriptive analysis and cross tabulations whereas STATA 14 was used for logistic regression analysis. Continuous data were summarized using descriptive statistics, whilst associations and linkages in categorical variables were evaluated using chi-square tests. To find predictors of antibiotic use, logistic regression analysis was utilized.

Ethical clearance to conduct this research was obtained from the Health Research Ethics Committee of the University of the Witwatersrand, Johannesburg South Africa (HREC. Protocol Number M190412); the Kenyatta National Hospital and University of Nairobi Ethics and Research Committee (KNH/UoN-ERC. Ref. No. P71910/2018); and the National Commission for Science and Technology and Innovation. Ref No. NACOSTI/P/19/3232/28732). To protect privacy, study areas were coded. 

## 3. Results

### 3.1. Participant Information

In the five informal communities, 447 households were investigated. Of these, 75% (n = 337) of responders were female, whereas 25% (n = 110) were male. Table 1 shows the sociodemographic characteristics of the participants, including their gender, age, place of residence, and HIV status. The majority of the female responders were under the age of 45 (79%). The respondents’ minimum age was 24 and their maximum age was 80. Manyatta A had the most participants (32.1%), followed by Nyalenda A (21.4%), Nyalenda B (21.2%), Manyatta B (19.9%), and Obunga (5.4%). As HIV affects the immune system, making people more susceptible to infections and, as a result, to increased antibiotic use, we needed to know the HIV status of the respondents; only around 5% of the participants were HIV positive. 

### 3.2. Prevalence of Antibiotic Use

Antibiotic use in the selected households during the previous month from the day of the interview was reported to be 43%. (193). Among individuals who consumed antibiotics, 70% (n = 137) reported getting the antibiotic through a prescription, whereas the rest did not consult a doctor for the antibiotic prescription. Further examination in the households that reported self-medication revealed that antibiotics were either used based on previous experience (76%) or on the advice of friends (26%). 

Additionally, the level of awareness of the health implications of drinking antibiotic-contaminated water was evaluated and it was discovered that only 35% (n = 158; *p* = 0.003) of households who reported using antibiotics were aware of the health risks of drinking antibiotic-contaminated water. Table 2 summarizes an in-depth examination of the distribution of antibiotic sources based on the characteristics of the respondents. Remarkably, a large proportion (87.5%) of HIV + respondents received a prescription from a medical practitioner for antibiotics taken. In terms of different settlement areas, a relatively small percentage of respondents in Manyatta B obtained an antibiotic prescription from a medical practitioner, which could explain the substantial difference seen (*p* < 0.000) among informal settlements. 

Furthermore, to comprehend the amount of health consciousness The odds of employing antibiotics were calculated using a 95% confidence interval, and no statistically significant difference was observed (Table 3). Female individuals had higher odds of using antibiotics than male participants, with an odds ratio of 1.33; z = 1.24, SE = 0.3105, *p* > 0.05. Interestingly, the odds of using antibiotics were shown to be lower among HIV-positive participants than among HIV-negative participants (0.313; z = −1.73, SE = 0.2097, *p* > 0.05). Participants aged 45 years and older showed a higher likelihood of using antibiotics compared to younger participants (odds ratio = 1.54, z = 1.22, SE = 0.550, *p* > 0.05). In terms of geographical location, the situation differs among different settlements, with the use of antibiotics decreasing in only one informal settlement (Nyalenda B) (odds ratio =0.42). z = 1.07, SEM = 1.726156, the observations were judged to be within one standard deviation of the mean.

### 3.3. Antibiotic Disposal

Proper disposal of leftover antibiotics is regarded as a major concern and has been linked to the development of antibiotic resistance. In this study, all participants were interviewed about the various methods they disposed of leftover medicines. Respondents disposed of antibiotics in a variety of methods, including 51.6% (n = 16) in pit latrines, 16.1% (n = 5) in compost pits, and 6.5% (n = 2) by burning. However, the majority of respondents (87.1%; n = 27) did not discard the unfinished antibiotics but preserved them for future use (Table 4). Antibiotic completion analysis revealed that females completed their antibiotic doses more than males.

A statistically significant difference in HIV status was observed, (*p* = 0.000), with the majority of HIV-positive respondents completing their antibiotic dose and just a tiny fraction of HIV-negative respondents not completing their antibiotic dose. There was also a difference among the informal settlements (*p* = 0.001), with a lower proportion of responders in Nyalenda B completing their antibiotic doses than in the other settlements. In terms of age group variations, it was discovered that antibiotic completion rose with age, although there were no significant differences in antibiotic dose completion across the various age groups (Table 4). 

### 3.4. Ground Water Use

Since improper disposal of unfinished antibiotics contaminates groundwater, it was critical to examine groundwater use in the selected settlements. As expected, more than 99% of respondents reported using groundwater for various household reasons (Table 5). While the majority of groundwater was reported to be used for washing clothes (96%), cleaning houses (95%), and washing utensils (78%), the same water was also seen to be utilized for drinking (9%) and cooking (18%) purposes. In terms of gender, a greater proportion of male respondents (12.6%) reported drinking groundwater than the opposite sex. HIV-positive subjects (19%) reported drinking from groundwater, but HIV-negative participants (8.9%) did not. Residents of the Manyatta B and Nyalenda B informal settlements were more likely to use groundwater for drinking than residents of the other settlements. We discovered that younger people did not drink groundwater, but that drinking groundwater increased with age. As water use is ubiquitous, no significant difference was found when comparing participant characteristics. 

## 4. Discussion

Lack of accurate diagnosis, combined with self-medication, is a rising global problem that leads to antibiotic abuse, particularly in low- and middle-income countries. Antibiotic use was reported to be 43% among Kisumu County residents living in informal settlements. However, an entry and exit survey on antibiotic use in an urban informal settlement discovered a high prevalence of antibiotic use at 87% (entry) and 70% (exit) [14]. Antibiotic use in these informal settlements is higher than the level reported in Europe in 2016, where 34% were reported to have used antibiotics [40]. In this study, the majority (70%) of antibiotic users in informal settlements obtained the medication on a prescription, but some also obtained the medication after seeking advice from friends for advice. Other participants utilized leftover antibiotics from a previous experience and some obtained the antibiotics after doing an online search for the symptoms and going directly to a chemist. This is comparable to a comprehensive investigation conducted in Uganda, where it was revealed that self-medication is common in low-income countries, with an overall prevalence of 38.8% for different ailments [41]. Self-medication was viewed as more convenient, less expensive, and less time-consuming than getting a prescription from a medical facility [15]. Antibiotics were saved for future use from a variety of sources, including pharmacies, leftover doses, and drug sharing, with education, age, gender, previous successful usage, the severity of illness, and income identified as factors influencing the behavior [15,41]. In another study, despite legal requirements that antibiotics be dispensed only with a prescription, antibiotics were obtained without a prescription, owing to increased access and poor enforcement of the legislation [15]. Antibiotic overuse is exacerbated by the ease with which antibiotics can be obtained without a prescription; however, there is a scarcity of information on antibiotic use and/or misuse in Kenya [17]. This study did not determine whether the correct dosage was administered in order to determine dosage compliance, which is an important factor that should be investigated further.

Self-medication with antibiotics is a major contributor to the development of drug resistance [41] and is typically conducted for the user’s convenience, resulting in antibiotic misuse [35]. Self-medication was found to be common in southern and eastern European countries with high levels of antibiotic resistance [42]. It has also been observed that the availability of leftover antibiotics is a key contributor to the habit of self-medication [43]. There is a need to develop a rigorous data collection system for antibiotic usage as well as to raise awareness about the health implications of antibiotic exposure in the environment, including groundwater.

An estimated 60% of Kisumu inhabitants live in informal settlements, where they face deficiencies such as a lack of clean water and basic sanitation [38]. Groundwater is utilized for household reasons by 99.8% of the people in these settlements, including cooking, drinking, washing utensils, washing clothes, and cleaning the house. This study supports the findings of a review of groundwater used for drinking in south-east Asia and the Pacific, which found that the average proportion of families using groundwater for drinking is high (66% (range of 22–95%) in urban areas and 60% (range of 17–93%) in rural regions) [44]. Okotto et al. (2015) discovered that an estimated 472m^3^ of groundwater was extracted but was used for other domestic purposes than drinking and cooking in the Kisumu informal settlements [45]. Another study in Kiambu County, Kenya, investigated the extent of groundwater use for domestic and irrigation purposes and discovered that residents perceived groundwater to be contaminated, with only 36.7 percent of households using groundwater for domestic purposes and 13.3% using it for irrigation [46].

The improper disposal of unused, stored, and expired antibiotics is of major concern [47], as users may be unaware of suitable disposal measures [48]. Antibiotics contaminate groundwater in a variety of ways, including direct dumping with rubbish, faeces and urine excretion, and hospital and industrial waste [1,28,49,50]. The same pattern was found in this study, where respondents reported disposing of antibiotics in pit latrines, compost pits, or burning. Pit latrines have been reported to be the most commonly used form of sanitation in facilities in informal communities [51]. However, other causes of water contamination have been identified, including industrial, agricultural, and flooding [46], which create calamity when it rains owing to the study area’s shallow water table [29]. However, several respondents did not discard the antibiotic but saved it for future use. Antibiotic usage has been linked to an increase in the likelihood of incorrect disposal into the environment [52]. Keeping the remainder of antibiotics for use the next time a person becomes ill is risky since there is a chance that the antibiotics will expire. The efficacy and potency of the antibiotics are compromised and are bound to fail to treat the intended infection [53]. They are also harmful and likely to lead to toxicity that may lead to irreversible damage to body organs [54]. Unfortunately, antibiotic resistance caused by poor antibiotic disposal is hardly addressed and is not a priority even among healthcare practitioners [52]. The literature has demonstrated that contamination of groundwater sources is higher in places with a high population density [55], which is a typical feature in informal settlements, but information on antibiotic usage and disposal in informal settlements has not been documented.

People with weakened immune systems use antibiotics the most to combat opportunistic infections. In this study, the self-reported HIV status was 4.7%, with a higher likelihood of utilizing antibiotics and completing their antibiotic dosage. This study’s location is in a region in Kenya where the HIV prevalence is greater than the national prevalence, which has been reported to be between 16.3% [56] and 17.5% [57]. An estimated 60% of Kisumu inhabitants live in informal settlements, where they face deficiencies such as a lack of access to services such as clean water and good sanitation [38]. The environment harbours resources such as groundwater, which serves as a complement to intermittent water supply for residential use, and it is critical that initiatives aimed at educating people living in informal settlements about the dangers of consuming unclean water are implemented [58]. As a result of knowing the HIV burden in the study region, knowing it follows that protecting the environment from antibiotic contamination is a key step in promoting human well-being in the one health setting. This study describes findings from a Kenyan informal settlement; however, it is well known that informal communities around the world face similar difficulties of access to essential amenities. The findings indicate that countries have a responsibility to carry out the global antimicrobial resistance action plan.

## 5. Implications for Policy and Areas for Further Research

Despite attempts to raise awareness about antibiotic usage and resistance through worldwide annual events, consumers still lack reliable knowledge about proper antibiotic use. To reduce misuse and abuse, it is critically important to design methods that assure measurement of antibiotic consumption by countries and use thereof at individual and household levels.

More research on prescribers’ and users’ knowledge of the amount of use and health impacts as a result of antibiotic exposure in the environment owing to inadequate disposal, should be conducted. It is also necessary to devote time to improving understanding through surveillance and study. These measures will reduce the disease burden and promote well-being, particularly in informal settlements, which have similar cross-cutting inequities.

Antibiotic disposal practices in informal communities have been found to pollute the environment, including groundwater; the result being environmentally induced antibiotic resistance. Antibiotic use reduction and correct disposal are strategies for combating antibiotic resistance in groundwater. Governments, as well as other relevant organizations and stakeholders, should prioritize the provision of safe water for all, particularly disadvantaged residents of informal settlements. Proper disposal techniques must be prioritized, and hence raising awareness about these among prescribers, providers, and antibiotic consumers should be a top concern.

### Limitations

Since this was an academic study with restricted resources, a large sample could not be studied. However, the study’s findings are applicable to and can be generalized to similar settings, because informal settlements have traits that are not necessarily unique to a particular geographical area. To look for differences between informal settlements, a different study approach than descriptive is recommended. For reason that the study was dependent on the response given, the actual HIV status of the respondents may not have been captured. A study targeting HIV patients at the clinic could be a better measure of the burden of disease among the informal settlements, and thus more concrete conclusions could be made on antibiotic use among HIV patients.

## 6. Conclusions

In Kenya, there is little information on antibiotic use at the household level. This study discovered that antibiotic use is significantly high, with disposal procedures similar to those known to contribute to environmental contamination, including groundwater. There was antibiotic misuse and abuse, as well as improper disposal, along with a lack of awareness of the health risks of antibiotic exposure in the environment, such as drinking antibiotic-contaminated water. The majority of informal settlement dwellers rely on groundwater to fulfill their water demands. As a result, there is a need for studies to increase surveillance of antibiotic usage and disposal. Additionally, governments and other relevant organizations must strengthen their commitment to groundwater resource management in order to offer sustainable universal water distribution free from the inclusion of pollutants such as antibiotics, particularly among disadvantaged communities in informal settlements.

## Figures and Tables

**Table 1 ijerph-20-00287-t001:** Sociodemographic and clinical characteristics of the study participants.

Characteristic	Description	n (%)
**Sex**	Male	110 (25.0%)
	Female	375 (75.0%)
**Age**	20–24	65 (14.5%)
	25–34	173 (38.6%)
	35–44	115 (25.7%)
	45+	94 (21.2%)
**Settlement**	OBUNGA	24(5.4%)
	MANYATTA A	143 (32.1%)
	MANYATTA B	89 (19.9%)
	NYALENDA A	96 (21.4%)
	NYALENDA B	95 (21.2%)
**HIV status**	Positive	21(4.7%)
	Negative	398 (88.8%)

**Table 2 ijerph-20-00287-t002:** Percentage distribution of sources of antibiotics used against demographic and clinical characteristics of the study participants.

Characteristics		Source of Antibiotic Used					
		Prescribed by a Doctor % (n)	Advice from Friends % (n)	Previous Experience % (n)	Online Search % (n)	Other (Chemist) % (n)	*p*-Value
**Sex**	Male	77.1 (37)	40.0 (4)	80.0(8)	0.0(0)	1.7(2)	0.408
	Female	73.8(107)	24.3(9)	75.7(28)	2.7(1)	1.2(4)	
**HIV status**	Positive	87.5 (14)	0 (0)	100 (2)	0(0)	0 (0)	0.001
	Negative	75.2(121)	31.6(12)	76.3(29)	2.6(1)	1.3 (5)	
**Settlements**	OBUNGA	81.8(9)	0 (0)	50 (1)	0 (0)	4.2 (1)	0.000
	MANYATTA A	81.6(62)	0 (0)	100 (13)	0 (0)	0 (0)	
	MANYATTA B	64.9 (24)	69.(9)	84.6(11)	0 (0)	1.1 (1)	
	NYALENDA A	70.2 (33)	28.6 (4)	71.4 (10)	7.1 (1)	0 (0)	
	NYALENDA B	72.7 (16)	0 (0)	20.0(1)	0 (0)	4.2 (4)	
**Age**	20–24	70.8 (17)	71.4 (5)	28.6 (2)	0 (0)	4.6 (3)	0.086
	25–34	72.7 (48)	77.8 (14)	83.3 (15)	5.6 (1)	1.2 (2)	
	35–44	73.6 (39)	69.2 (9)	100 (13)	0 (0)	0 (0)	
	45+	80.0 (40)	66.7 (6)	66.7 (6)	0 (0)	1.1 (1)	

**Table 3 ijerph-20-00287-t003:** Predictors of antibiotic use.

Antibiotic Use		Odds Ratio	Std. Error	z	*p* > z	95% Conf. Interval
**Sex**	Female	1.33	0.3105	1.24	0.215	0.845–2.105
**HIV Status**	Positive	0.313	0.2097	−1.73	0.083	0.084–1.163
**Age (years)**	25–34	0.96	0.302	−0.12	0.908	0.522–1.782
	35–44	0.99	0.335	−0.03	0.980	0.511–1.921
	45+	1.54	0.550	1.22	0.223	0.768–3.104
**Settlements**	MN A	1.54	0.721	0.91	0.362	0.611–3.855
	MN B	1.17	0.571	0.32	0.748	0.449–3.042
	NY A	1.61	0.778	0.98	0.325	0.623–4.153
	NY B	0.42	0.207	−1.76	0.078	0.157–1.103
**Cons**		2.263088	1.726156	1.07	0.284	0.507–10.091

OBG—Obunga; MN A—Manyatta A; MN B—Manyatta B; NY A—Nyalenda A; NY B—Nyalenda B.

**Table 4 ijerph-20-00287-t004:** Distribution of completion of antibiotic dose by demographic characteristics.

Characteristics	Description	Yes	No	*p*-Value
**Sex**	Male	39 (32.5)	7 (5.8)	0.647
	Female	117 (36.3)	27 (8.4)	
**HIV status**	Positive	15 (71.4)	1 (4.8)	0.000
	Negative	130 (32.7)	26 (6.5)	
**Settlements**	MN A	63 (43.8)	13 (9.0)	0.001
	MN B	30 (33.7)	6 (6.7)	
	NY A	35 (36.5)	11(11.5)	
	NY B	17 (17.9)	4 (4.2)	
	OBG	11 (45.8)	0 (0)	
**Age**	20–24	21 (32.3)	2 (3.1)	0.089
	25–34	55 (31.8)	10 (5.8)	
	35–44	40 (34.8)	11 (9.6)	
	45+	40 (42.6)	11 (11.7)	

OBG—Obunga; MN A—Manyatta A; MN B—Manyatta B; NY A—Nyalenda A; NY B—Nyalenda B.

**Table 5 ijerph-20-00287-t005:** Percentage distribution of groundwater use against sociodemographic characteristics.

Characteristics		Groundwater Use					
		Drinking n (%)	Cooking n (%)	Washing Utensils n (%)	Washing Clothes n (%)	House Cleaningn (%)	*p*-Value
**Sex**	Male	12.6 (15)	21.8 (26)	82.2 (97)	94.9 (112)	94.9 (112)	0.261
	Female	7.5 (24)	17.5 (56)	79.1 (253)	96.6 (309)	95.3 (303)	
**HIV status**	Positive	19.0 (4)	28.6 (6)	85.7 (18)	100.0 (21)	95.2 (20)	0.950
	Negative	8.9 (35)	18.0 (71)	79.2 (312) (79.2)	95.9 (378)	95.9 (376)	
**Settlements**	OBG	12.5 (3)	16.7 (4)	62.5 (15)	95.8 (23)	95.8 (23)	0.716
	MN A	1.4 (2)	9.9 (14)	92.2 (130)	100.0 (141)	92.8 (129)	
	MN B	13.5 (12)	25.8 (23)	74.2 (66)	93.3 (83)	95.5 (85)	
	NY A	10.5 (10)	20.0 (19)	80.0 (76)	97.9 (93)	95.8 (91)	
	NY B	13.7 (13)	24.2 (23)	69.5 (66)	91.6 (87)	97.9 (93)	
**Age**	20–24	3.1 (2)	12.3 (8)	72.3 (47)	95.4 (62)	90.8 (59)	0.660
	25–34	7.1 (12)	14.7 (25)	71.2 (121)	94.7 (161)	95.9 (163)	
	35–44	9.6 (11)	23.5 (27)	89.6 (103)	99.1 (114)	94.7 (107)	
	45+	16.0 (15)	24.5 (23)	87.1 (81)	95.7 (89)	97.8 (91)	

OBG—Obunga; MN A—Manyatta A; MN B—Manyatta B; NY A—Nyalenda A; NY B—Nyalenda B.

## Data Availability

Data are available on request due to the privacy of the respondents. The data presented in this study are available on request from the corresponding author. Data are not publicly available due since this is an academic study and some of the data are still being analyzed for publication.

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
