# Peer review of "Prevalence of Antibiotic Use and Disposal at Household Level in Informal Settlements of Kisumu, Kenya"

_ijerph, 2022, doi:10.3390/ijerph20010287_

Round 1
Reviewer 1 Report (New Reviewer)
In addition to the reported use of the antibiotics, it would be interesting to be examined in how many of the cases the use was justified, the antibiotic was the appropriate and in the correct dose and with the appropriate duration of treatment. Of course in another study.
Author Response
Dear Sir/Madam,
Response: The study will be used as a point of reference when conducting antibiotic related studies in the region and globally. An investigation of the actual antibiotic consumed, the appropriateness and correctness of the dosage for the suitable period of treatment is necessary. This will require a form of funding which I can apply for and implement.
Kind regards,
Kellen
Reviewer 2 Report (New Reviewer)
Reviewer Report
Manuscript ID: ijerph-2015736 entitled” Prevalence of Antibiotic use and disposal at household level in informal settlements of Kisumu, Kenya.”
Comments to the Authors
I have the following comments:
Ø It is found that 54% of respondents did not always complete doses but kept the remainder for next time they would get sick. About 32% of the remainder of antibiotics were disposed in pit latrines, compost pits (10%) while 4% disposed of by burning. The total becomes exactly 100%. What about expired drugs? When they keep the drugs for next use there is a maximum chance that some of the drugs will expire. Please discuss this in the discussion section.
Ø If there is non compliance of drugs and dispensing without prescription, there may also be dispensing as per requirement e.g. 6 tablets in place of 10 tablets. There is no mention of this in the discussion section of the article.
Ø In table 3, 4 and 5 the full forms of OBG, MN A, NY A etc could be added below the table.
Ø The language can be further improved.There are many grammatical mistakes, and the English language usage and grammar need to be checked and improved by native English.
Author Response
Dear Sir/Madam,
Thank you for the valuable reviews. They have helped me improve the document.
Please find attached responses to the comments.
Kind regards,
Kellen

Reviewer 3 Report (New Reviewer)
Dear Editor
In this paper, authors investigated about improper disposal in informal settlements as a source of contamination in the environment that include groundwater, and determined antibiotic use and disposal in informal settlements of Kisumu, Kenya. As a result, misuse and inappropriate disposal of antibiotics identified may lead to higher risk antibiotic resistance. The emergence of resistant bacteria has become a problem worldwide, and it is very significant to investigate the environmental impact of not only the use but also disposal of antibiotics. This study is a questionnaire survey, and does not investigate the actual type of antibiotic or its concentration in the environment, and it does not show academic impact. In addition, the same content can be seen in the text, and there are many grammatical mistakes. The impact of the findings obtained from this study is also low, and the sample size is small.
Although this paper contains very significant findings, it is not possible to read the academic impact and novelty, and the text needs to be reviewed. Based on the above points, I judged that it would be difficult to accept.
Sincely yours,
Author Response
Dear Sir/Madam,
Thank you for your valuable comments. They have helped me improve my document.
Please find attached responses to the comments
Kind regards,
Kellen

Round 2
Reviewer 2 Report (New Reviewer)
Reviewer Report
Manuscript ID: ijerph-2015736 R1 entitled” Prevalence of Antibiotic use and disposal at household level in informal settlements of Kisumu, Kenya.”
Comments to the Authors
I have the following comments:
Authirs are justified our comments still some of minor revision of MS should be required
Inappropriate the citations made by the authors; insufficent methodology are found in the MS and also Native language can be further improved.There are many grammatical mistakes, and the English language usage and grammar need to be checked

Reviewer 3 Report (New Reviewer)
I think this paper is acceptable in present form.
This manuscript is a resubmission of an earlier submission. The following is a list of the peer review reports and author responses from that submission.
Round 1
Reviewer 1 Report
Dear Authors. It was interesting to read your work on antibiotic use and disposal in informal settlements. In your questionnaire, you did not assess sources of water for different households, yet you assessed knowledge on the consumption of antibiotic-contaminated water. This creates a missing gap in your data.
Again you might need to clarify how the households would know that water is contaminated or not and associated health hazards.
What are other sources of contamination of water other than household disposal
Author Response
REVIEWER 1
Comment 1: Dear Authors. It was interesting to read your work on antibiotic use and disposal in informal settlements. In your questionnaire, you did not assess sources of water for different households, yet you assessed knowledge on the consumption of antibiotic-contaminated water. This creates a missing gap in your data.
Response 1: Data on groundwater use in the informal settlements has been included. –Page 8 and 9
Comment 2: Again you might need to clarify how the households would know that water is contaminated or not and associated health hazards.
Response 2: Practices of making water safe for drinking like boiling or adding chlorine are a common among residents of the informal settlements. For our case the question of if they were aware of effects of drinking water that has been contaminated with antibiotics
Comment 3: What are other sources of contamination of water other than household disposal
Response 3: The other sources of contamination other than household sanitation facilities and disposal have been identified to be industrial waste, agricultural contamination and flooding- Page 4
|
Yes |
Can be improved |
Must be improved |
Not applicable |
|
|
Does the introduction provide sufficient background and include all relevant references? |
(x) |
( ) |
( ) |
( ) |
|
Are all the cited references relevant to the research? |
(x) |
( ) |
( ) |
( ) |
|
Is the research design appropriate? |
( ) |
(x) |
( ) |
( ) |
|
Are the methods adequately described? |
( ) |
(x) |
( ) |
( ) |
|
Are the results clearly presented? |
(x) |
( ) |
( ) |
( ) |
|
Are the conclusions supported by the results? |
(x) |
( ) |
( ) |
( ) |
*The research design has been elaborated and now clear
*The methods section has been adequately described
Reviewer 2 Report
This is just a small questionnaire based case study; no empirical evidence produced in support of the conclusion, No important result reported.
Reviewer 3 Report
The manuscript entitled “
Prevalence of Antibiotic use and disposal at household level in 2 informal settlements of Kisumu, Kenya.” by Karimi et al. talks about the reckless use of antibiotics use and its impact on a specific group of people. Although the work has potentially good information, however, the work is more of a survey based than an actual scientific study.
The work is very specific to a geographical region and may not apply to wider public health.
The work does not contain any significant scientific data to propagate this important field of research. I would recommend it to publish it in a more specific journal than the international journal of environmental research and public health